# Phylogenetics and genomic variation of *Hepatocystis* isolated from shotgun sequencing of wild primate hosts

**Paige E. Haffener**[ID], **Helena D. Hopson, Alejandra Herbert-Mainero, Anayansi Ramirez, Ellen M. Leffler**[ID]*

Department of Human Genetics, The University of Utah School of Medicine, Salt Lake City, Utah, United States of America

* leffler@genetics.utah.edu

## Abstract

*Hepatocystis* are apicomplexan parasites nested within the *Plasmodium* genus that infect primates and other vertebrates, yet few isolates have been genetically characterized. Using taxonomic classification and mapping characteristics, we searched for *Hepatocystis* infections within publicly available, blood-derived whole genome sequence (WGS) data from 326 wild non-human primates (NHPs) in 17 genera. We identified 37 *Hepatocystis* infections in *Papio cynocephalus* (yellow baboons) and four species of *Chlorocebus* monkeys (grivets, green monkeys, vervet monkeys, and malbroucks) sampled from locations in west, east, and south Africa. *Hepatocystis cytb* sequences from *Papio* and *Chlorocebus* hosts each clustered within host species among previously reported isolates from other NHP taxa. Utilizing the low-coverage sequence data (0.11-0.76X per sample) recovered across the nuclear *Hepatocystis* genome, we identified 349,893 polymorphic sites. Principle components analysis based on genotype likelihoods across all samples showed evidence for population structure by primate host species. Across the genome, windows of high SNP density revealed candidate hypervariable loci including *Hepatocystis*-specific gene families possibly involved in immune evasion and genes that may be involved in adaptation to their insect vector and hepatocyte invasion. Overall, this work demonstrates how WGS data from wild NHPs can be leveraged to study the evolution of apicomplexan parasites and potentially test for association between host genetic variation and parasite infection.

## Author summary

Non-human primates are hosts to many species of *Plasmodium*, the parasites that cause malaria, and a closely related group of parasites called *Hepatocystis*. However, due to restrictions and challenges of sampling from wild populations,

**Data availability statement:** All non-human primate datasets used in this analysis were publicly availably NCBI BioProjects, and accession numbers can be found in S1 Table. BAM files of reads mapped to Hepatocystis as well as major and minor allele calls for cHEP and pHEP are available at zenodo: 10.5281/zenodo.12209843.

**Funding:** This work was supported by the United States National Institutes of Health (grant number R35 GM147709 to E.M.L.). P.E.H. was supported by T32 GM007464 and T32 GM141848. The funders had no role in study design, data collection and analysis, decision to publish, or preparation of the manuscript.

**Competing interests:** The authors have declared that no competing interests exist.

we lack a complete understanding of the breadth of diversity and distribution of these parasites. Here, we provide a framework for testing already-sampled populations for parasite infections using whole genome sequences derived from whole blood samples from the host. Following taxonomic classification of sequences from 326 wild primates using a database of reference genomes, we mapped reads to candidate parasite genomes and used nuclear and mtDNA coverage metrics to identify infections. Through this approach, we identified 37 *Hepatocystis* infections in two genera of African old world monkeys (*Papio* baboons and *Chlorocebus* monkeys). Investigating genetic variation across the *Hepatocystis* genome, we found that parasite populations are structured by primate host, and we also described genes that may be under immune selection. This approach can be applied to additional sequencing datasets from non-human primates and other vertebrate hosts as well as datasets from invertebrate vectors to improve our understanding of where these parasites are found, their host-specificity, and their evolutionary history. This framework may also be adapted to study host-pathogen evolution in other systems.

## Introduction

Despite maintaining a distinct genus name, *Hepatocystis* parasites cluster phylogenetically within the *Plasmodium* genus as a sister clade to the rodent malaria parasites [1–3], which serve as model systems for human malaria [4]. Like *Plasmodium*, the *Hepatocystis* life cycle includes infection of hepatocytes followed by erythrocytes and transmission through a vector blood meal [5,6]. However, they lack several key characteristics of *Plasmodium* parasites. Notably, *Hepatocystis* do not undergo asexual replication in red blood cells, when symptoms of malaria occur, and are transmitted by midges of the *Culicoides* genus rather than by *Anopheles* mosquitoes [2,3,7]. *Hepatocystis* parasites infect a wide range of vertebrate hosts including non-human primates (NHPs), although they are not thought to infect humans [1,6]. Surveys of *Hepatocystis* species diversity have been conducted in bats [8–16] and NHPs, but are more limited in the latter in part due to difficulty of obtaining blood samples from wild individuals.

In NHPs, six species of *Hepatocystis* have been described via microscopy, all in old world monkeys (OWMs) [1]. In South Asia, two *Hepatocystis* species have been reported in the genus *Macaca*: *H. semnopitheci* in long-tailed and pig-tailed macaques in southern Thailand and *H. taiwanensis* in Formosan rock-macaques in Taiwan [6,17]. Genetic surveys of *cytochrome b* (*cytb*) indicate that *Hepatocystis* is prevalent in Thai macaques (44–55% infected), but the sequence data has generally not been linked to a morphologically described species [18]. The remaining four morphologically described species of *Hepatocystis* - *H. kochi*, *H. bouillezi*, *H. simiae*, and *H. cercopitheci* - have been found in multiple genera of African OWMs (*Cercopithecus*, *Cercocebus*, *Chlorocebus*, *Colobus*, and *Papio*) [6,19–22]. Infection prevalence, determined by either microscopy or mtDNA sequencing, ranges from 0% to over 60%

in populations of OWMs from Cameroon, Uganda, Tanzania, Kenya, and Ethiopia [20–25]. *Hepatocystis* mtDNA has been reported in one human fecal sample as well as fecal samples from chimpanzees (*Pan troglodytes schweinfurthii*, *Pan troglodytes elioti*, and *Pan troglodytes troglodytes*) in Uganda, Cameroon, the Democratic Republic of Congo, and Tanzania, suggesting *Hepatocystis* may also be able to infect some great apes [25]. However, a dietary source could explain *Hepatocystis* sequences in fecal samples, and *Hepatocystis* DNA has so far only been recovered from a single chimpanzee blood sample [25]. Phylogenetic trees of *Hepatocystis cytb* sequences isolated from different species of African OWMs and chimpanzees do not exhibit overt geographic clustering or host-specificity [24–26]. Nonetheless, limited sampling, both across primates and the genome hinder a deeper understanding of co-evolution between *Hepatocystis* and their NHP hosts.

Although phylogenetic analysis of *cytb* is informative, genomic sequences are necessary to robustly infer evolutionary relationships and to gain insight into patterns of genetic variation and genome evolution. The first and only *Hepatocystis* spp. genome sequence was published in 2020 based on sequences from an infected red colobus monkey (*Piliocolobus tephrosceles*) [3]. The genome assembly revealed that several loci known to be involved in *Plasmodium* mosquito stages had a high non-synonymous substitution rate relative to *Plasmodium* and that several genes important in liver stages had increased copy numbers. These hint at possible adaptations *Hepatocystis* may have evolved for utilizing *Culicoides* spp. as a vector and primarily infecting hepatocytes, respectively [3]. Additionally, genes involved in erythrocytic schizogony were either found to be present with fewer copies than in *Plasmodium*, such as *pir* genes, or were entirely absent, including those encoding reticulocyte binding proteins [3]. While these results are groundbreaking in furthering our understanding of *Hepatocystis* evolution, they remain the only genomic sequence data for *Hepatocystis* spp. and reflect a single isolate.

In this work, we aim to expand on the existing phylogenetic and evolutionary analyses of *Hepatocystis* parasites by extracting *Hepatocystis* sequences from publicly available shotgun sequences of wild NHPs. We identify infections in *Papio cynocephalus* (yellow baboons) and four species of *Chlorocebus* monkeys (grivets, green monkeys, vervet monkeys, and malbroucks) from Ethiopia, Kenya, and South Africa, consistent with where infections have been reported in previous studies, as well as in Zambia and The Gambia, building on our knowledge of the range of *Hepatocystis* spp. Using both *cytb* and nuclear genetic variation obtained from both *Chlorocebus*- and *Papio*-infecting *Hepatocystis*, we find that isolates from each primate host cluster separately, suggesting parasite populations are structured by primate host. Finally, we identify genes showing high SNP density as candidates potentially undergoing diversifying selection in *Hepatocystis*.

## Results

### Curation of a sequence dataset from wild, non-human primates

To survey *Hepatocystis* and *Plasmodium* infections across NHP species and geographic locations, we searched the NCBI SRA database, a database of publicly available sequence reads, for shotgun sequence data from wild NHPs derived from blood samples. We identified 326 such samples from 17 different primate species originating from 18 countries across Africa, Asia, and the Caribbean (Fig 1 and S1 Table). In total, 83% of samples were from African OWMs, 3% from Asian OWMs, and 14% from great apes. Most samples (76%) were collected in Africa. Of the remaining samples, half were collected in the Caribbean Islands and half were collected in Asia.

### *Hepatocystis* infections in African old world monkeys

We searched for *Plasmodium*- or *Hepatocystis*-derived sequences in the read data for all 326 NHP samples using Kraken2. Kraken2 is a kmer-based taxonomic classification tool that compares unique kmers from sequence reads to numerous reference genomes [27,28]. We identified samples with an elevated proportion of reads classified as *Hepatocystis* from two genera: *Papio* and *Chlorocebus* (S1A Fig). For further evaluation, reads from each *Papio* and

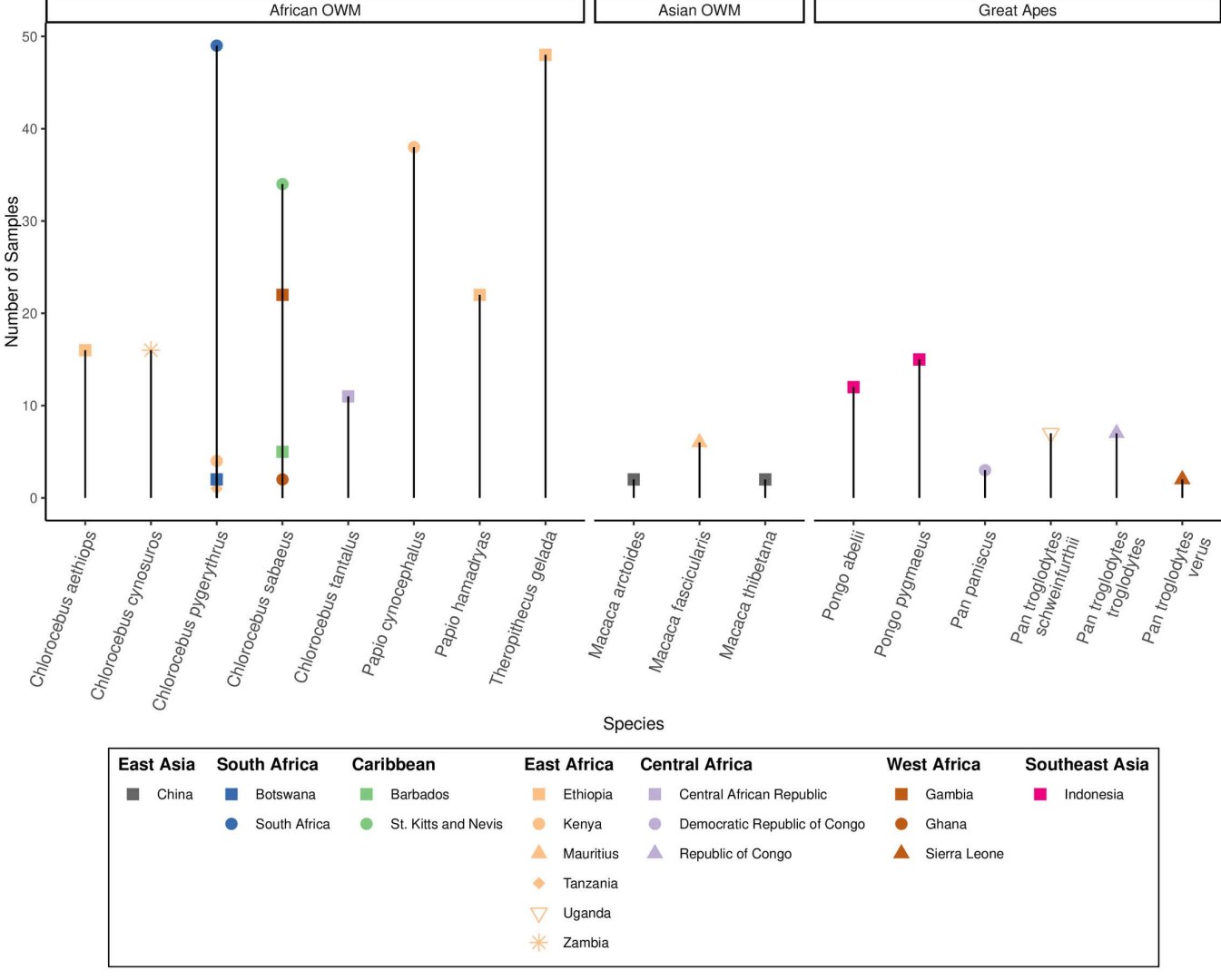

**Fig 1. Number of samples per species in the curated dataset of blood-derived whole genome sequence data from wild non-human primates.** Species are grouped by host genus and geography (OWM: Old World Monkey). Each colored point represents a collection of samples (BioProject) in the curated dataset. Points are colored by geographic region and shaped by country of sampling within each region.

*Chlorocebus* sample were mapped to a joint primate and *Hepatocystis* reference genome using bwa mem [29] (10.5281/zenodo.12209843). We assessed the distribution of coverage across nuclear and mitochondrial (mtDNA) contigs of the *Hepatocystis* reference (Fig 2A). We identified 37 samples meeting three criteria consistent with infection: (1) >0.002% of reads classified as *Hepatocystis*, representing 6 SD above the mean observed in a control analysis of human sequences from the French population in the Human Genomic Diversity Panel (presumed uninfected); (2) mtDNA coverage > 1X and (3) nuclear DNA coverage > 0.1X (Fig 2A and S2 Table). The infected samples include 23 individuals from four different species in the genus *Chlorocebus*: seven *C. aethiops* (grivets), three *C. cynosuros* (malbroucks), five *C. pygerythrus* (vervet monkeys) and eight *C. sabaeus* (green monkeys) as well as 14 *Papio cynocephalus* (yellow baboons), all sampled in Africa (Fig 2B and S3 Table). The average coverage was 0.31X and 0.26X for nuclear contigs and 26X and 16X for mtDNA across infected *Papio* and *Chlorocebus* samples, respectively. Although we applied the same pipeline to look for

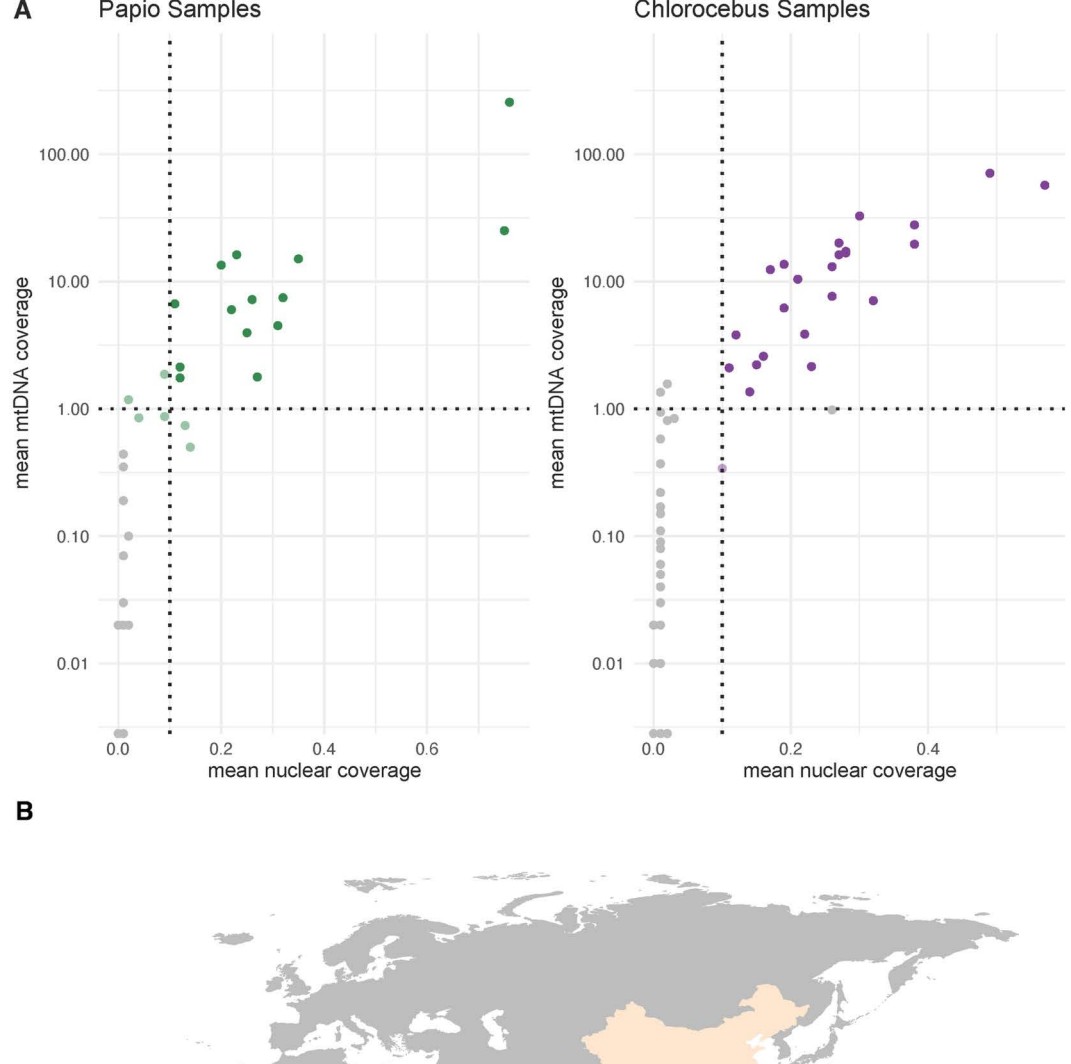

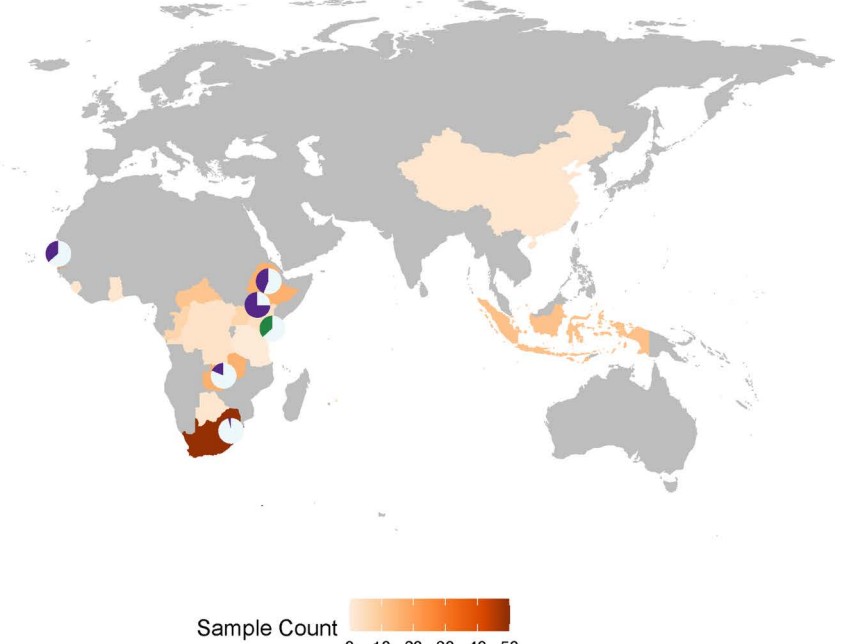

Sample Count
0  10  20  30  40  50

**Fig 2. Summary of infection inference.** A) Mean coverage of *Hepatocystis* nuclear and mitochondrial contigs for each *Papio* and *Chlorocebus* sample. Dark, colored points in the upper right quadrant indicate samples considered infected as they had mtDNA coverage > 1X, nuclear coverage > 0.1X and >0.002% of reads classified as *Hepatocystis* by Kraken2. Samples colored in a lighter shade outside this quadrant had > 0.002% reads classified as

*Hepatocystis* by Kraken2 but did not meet the coverage criteria. B) Map showing sampling locations for all 326 samples and infection prevalence in countries with where infected samples were found. In the pie charts, gray is the proportion uninfected and purple (*Chlorocebus*) and green (*Papio*) are the proportion infected. There were no samples from the countries colored gray. The world map was accessed in the *maps* package in R and is available from https://www.naturalearthdata.com.

*Plasmodium*, few samples showed an elevated proportion of *Plasmodium*-classified reads. In these, cases, mapping to a joint reference genome revealed coverage in only limited regions of the *Plasmodium* genome, suggesting it results from errors or other technical issues (S1B Fig).

To further support the inference of *Hepatocystis* in these samples, we examined the *Hepatocystis*-mapped reads. The average GC content of reads mapping to *Hepatocystis* in infected samples was 24.7% (range 23.8-26.7%), consistent with the expectation for this species. As expected with a true infection, sequence coverage was spread throughout the genome rather than on limited number of sites (S2 Fig). To rule out possible infections with *P. gonderi*, a *Plasmodium* species in the *vivax* clade that is known to infect these species but was not included in the Kraken2 database, we re-mapped the *Hepatocystis*-mapped reads, providing both the *Hepatocystis* spp. and *P. gonderi* reference genomes as mapping targets. In this competitive mapping, an average of 98.5% of the reads preferentially mapped to the *Hepatocystis* reference (range 96.7-99.4%). We conservatively removed reads that preferentially mapped to *P. gonderi* from further analysis.

### *cytB* phylogeny of NHP-infecting *Hepatocystis*

Given higher mtDNA sequencing depth and existing sequence data for comparison, we attempted to assemble *cytb* sequences from the 37 infected samples using SPAdes. This resulted in successful assembly of 20 *cytb* sequences, with 17 failing assembly either due to low or nonuniform coverage of the *cytb* region. The 20 cytb sequences include seven from *Papio cynocephalus* hosts (two unique) and 13 from *Chlorocebus* hosts (six unique). We combined our assembled *cytb* sequences and the reference sequence with 75 publicly available, unique *Hepatocystis cytb* sequences that have been identified in a range of NHPs as well as sequence from an Asian fruit bat (*Pteropus hypomelanus*) as an outgroup [17,18,23–25]. The resulting phylogeny shows the *Papio*-infecting sequences clustering with previously reported *Hepatocystis* sequences from *Papio* and *Piliocolobus* hosts, including the sequence in the reference genome (Fig 3). There are no previously-reported *Chlorocebus*-associated isolates, and those from our study cluster together and then with the single sequence from a *Colobus* host and several from *Cercopithecus* hosts (Fig 3). In *Chlorocebus aethiops* and *Chlorocebus sabaeus*, where we recovered multiple cytb sequences, sequences from the same host species were very similar. All five *cytb* sequences (three unique) from *C. sabaeus* hosts clustered together and the same *cytb* sequence was obtained from all six *C. aethiops* samples. We did not recover any identical *cytb* sequences from different primate host species. Given that different studies have sequenced different regions of *cytb*, we also constructed a phylogeny based on the sites present in all sequences in the alignment, yielding a similar topology (S3 Fig).

### Genomic variation in *Hepatocystis*

To assess patterns of genome-wide genetic variation, we focused on methods implemented in Analysis of Next Generation Sequencing Data (ANGSD [30]) that use genotype likelihoods, as this accounts for the individual genotype uncertainty in low-coverage population sequence data. We identified sites with evidence of single nucleotide polymorphism (SNP) variation across samples on nuclear DNA contigs, separately in *Hepatocystis* from *Papio* hosts (n = 14, mean coverage 0.31X, total coverage 4.3X) and from *Chlorocebus* hosts (n = 23, mean coverage 0.26X, total coverage 6.0X) as well as jointly (n = 37, total coverage 10.3X). Throughout the genome we identified 49,375 SNPs in *Hepatocystis* from *Papio* hosts (referred to from here on as pHep), 26,536 SNPs in *Hepatocystis* from *Chlorocebus* hosts (referred to from here on as cHep), and 349,893 SNPs when including all *Hepatocystis* from all hosts (10.5281/zenodo.12209843). Using these

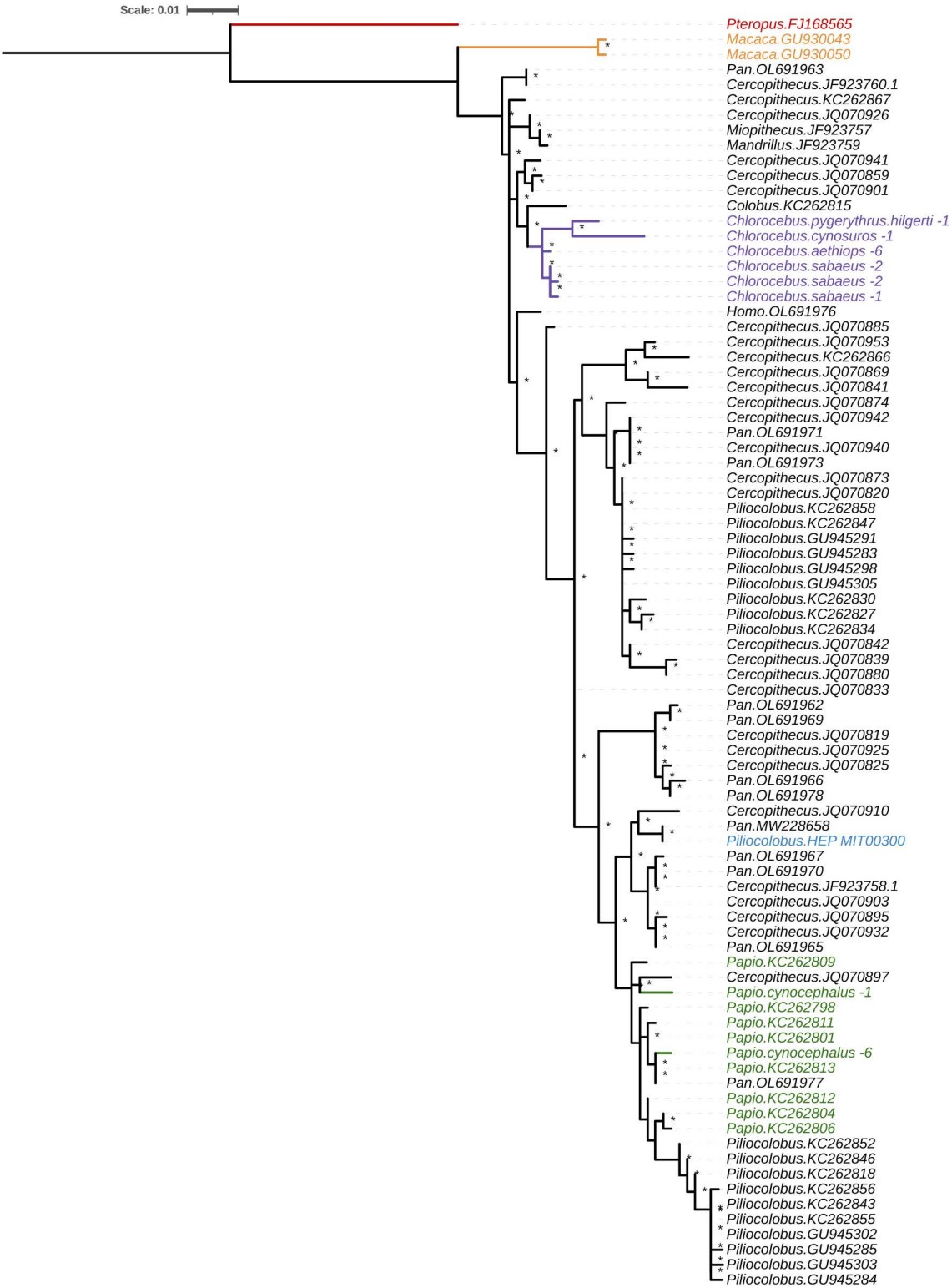

**Fig 3. Phylogenetic tree of _cytochrome b (cytb)_ sequences.** The maximum likelihood tree was constructed with IQ-TREE from the 20 _cytb_ sequence assemblies (eight unique) in our dataset together with the _Hepatocystis_ reference sequence and additional 75 unique publicly available _Hepatocystis_ sequences from various NHPs sequenced via PCR (alignment length = 727 bp, minimum sequence length = 494 bp, sites present in all

sequences = 287 bp). Tips representing sequences from our dataset are colored and labeled with the host species name and number of samples carrying the same sequence. Tips from published studies are labeled by host genus and a sample ID. Bootstrap values >50% are indicated with an asterisk. Color key: red – outgroup from bat host *Pteropus hypomelanus*. Purple – *Chlorocebus* hosts from our study. Dark green – *Papio* hosts. Light green – *Papio* hosts from other studies. Orange – Asian primates (macaques). Blue – the reference sequence. Black – African primates.

variants, we investigated parasite population structure (Fig 4) and calculated SNP density across the genome to identify hypervariable loci that may be involved in immune evasion in *Hepatocystis* (Fig 5).

**_Hepatocystis_ population structure in _Papio_ and _Chlorocebus_ hosts.** Although most individual genotypes cannot be confidently assigned, the overall genetic relationships between individuals can be inferred using the large number of SNPs across the genome by integrating over the genotype uncertainty [31,32]. Using the genotype likelihoods for SNPs across the genome from all 37 samples, we computed principal components using pcangsd [33]. In the combined SNP dataset including *Hepatocystis* from both pHep and cHep, the first PC clearly separates the 14 isolates from *Papio* from the 23 isolates from *Chlorocebus* hosts and accounts for a large proportion of the variance (Fig 4A). PCA computed within

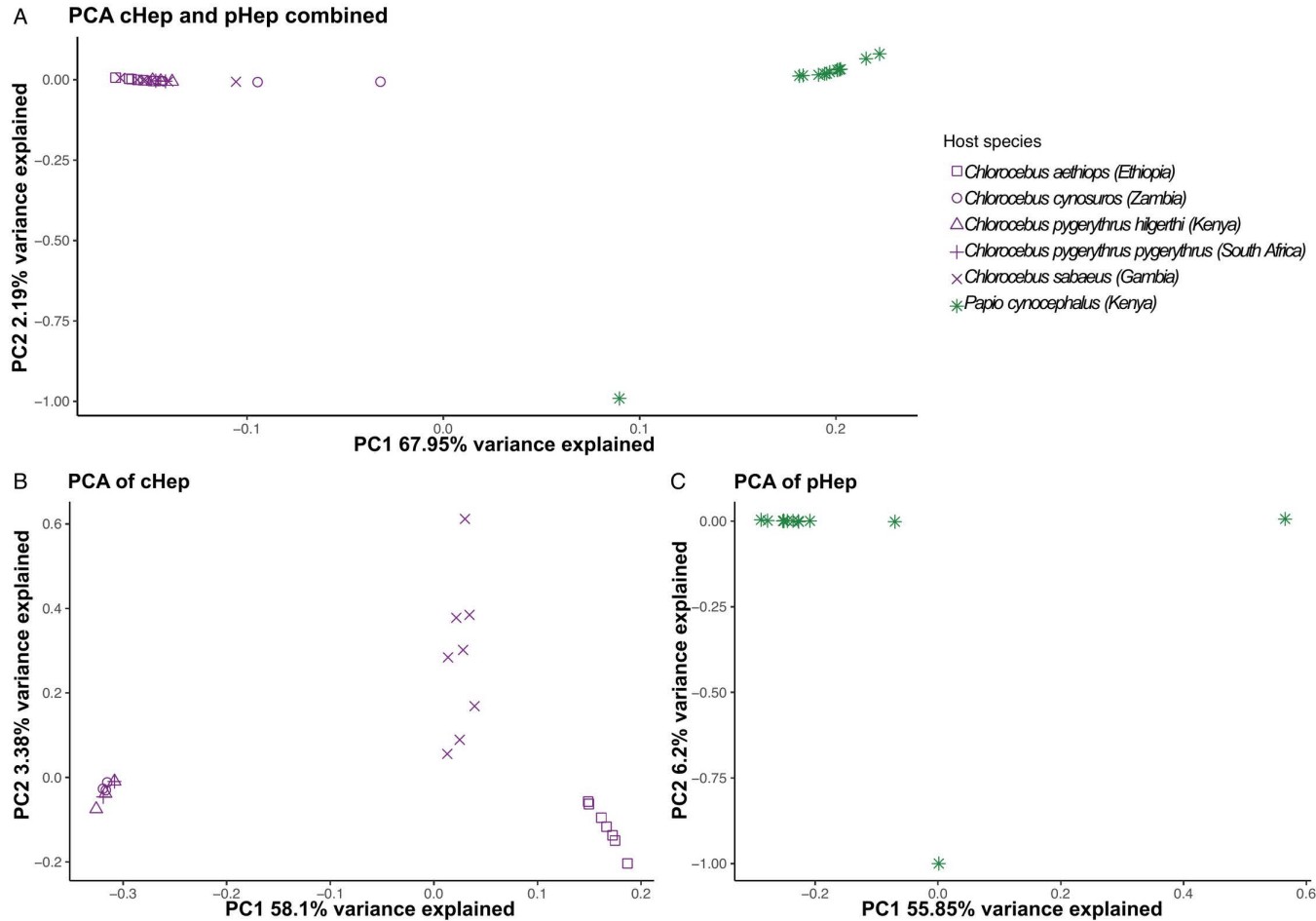

**Fig 4. PCA plots from genetic variation across the *Hepatocystis* nuclear genome.** A) First two principal components computed across both cHep and pHep. B) First two principal components computed for cHep. C) First two principal components computed for pHep. Shapes distinguish host species and location as indicated in the legend. Colors indicate host genus, with purple for *Chlorocebus* and green for *Papio*.

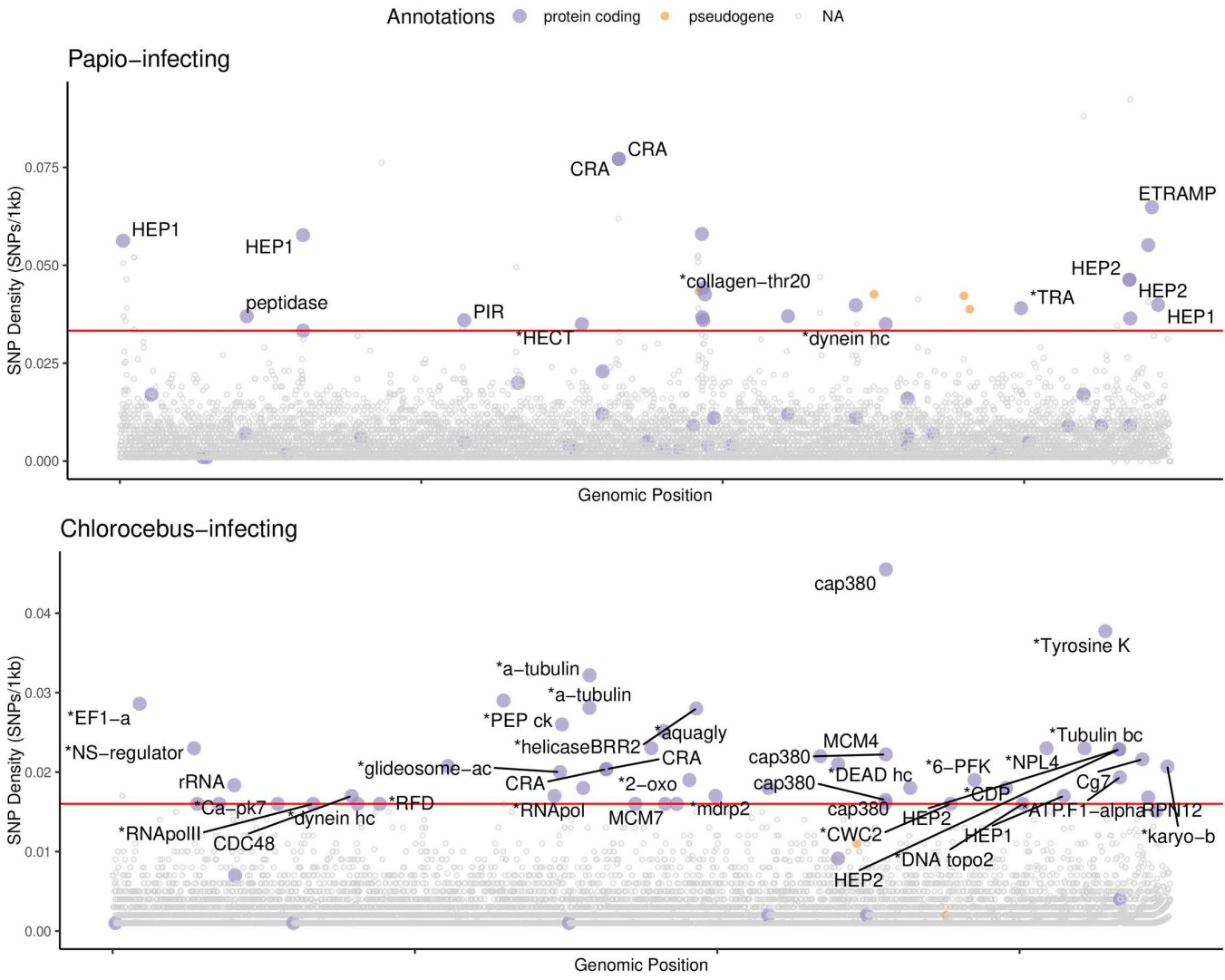

**Fig 5. SNP density in pHEP (top) and cHEP (bottom) in 1kb bins.** The x-axis is position along the *Hepatocystis* reference concatenating all contigs in order of decreasing size. In both plots, the solid red line represents the threshold for the 50 windows with the highest SNP densities. The top 50 points are annotated as follows: purple points indicate bins that overlap protein coding genes and are labeled with the gene name, if one has been assigned (* indicates a shortened name, with full names given in S5 and S6 Tables). Orange points overlap pseudogenes. Gray points above the line do not overlap a gene. Points in the top 50 of either pHep or cHEP are colored in both plots. Points not in the top 50 in either plot are gray.

cHep also shows clear separation between isolates from different *Chlorocebus* species, particularly distinguishing among parasites sampled from the *C. cynosuros/pygerythrus* group, *C. sabaeus,* and *C. aethiops* (Fig 4B). In pHep, which are sampled from a single *P. cynocephalus* population in Kenya, most samples form a single cluster, except for two samples, each separated on one of the first two PCs (Fig 4C). The outlier on PC2 corresponds to the parasite carrying a unique cytb haplotype (Fig 3), while the outlier on PC1 did not have a *cytb* assembly.

**Identification of hypervariable loci in *Hepatocystis*.** We tested our ability to recover hypervariable regions from low coverage data by applying the same pipeline to calculate SNP density in a *P. falciparum* dataset down sampled to 30X (378,737 SNPs), 1X (55,759 SNPs), 0.5X (26,281 SNPs) and 0.1X (1,117 SNPs) coverages. We used precision recall curves to determine a threshold for windows of high SNP density (S4 Fig). We considered a truth set

of hypervariable bins to be those in the 30X data with a SNP density greater than 2 standard deviations higher than the mean. Both 1X and 0.5X datasets had high precision (100% for the top 10 bins and >96% for the top 50 bins) and overlapped genes known to be hypervariable in *P. falciparum* (S4 Table). The 0.1X dataset did not perform as well, but the number of SNPs in this dataset for *P. falciparum* was extremely low, likely due to the much lower genetic diversity (~10x lower) than most other *Plasmodium* [34]. The number of SNPs in our dataset (26,536 for cHep and 49,375 for pHep) more closely match 0.5X and 1X for *P. falciparum*. We considered the top 50 bins in our dataset as candidate hypervariable regions.

Protein-coding genes (and not pseudogene members) from the *Hepatocystis*-specific *Hep1* and *Hep2* gene families were among the most hypervariable in both pHep and cHep (Fig 5 and S5 and S6 Tables). Both gene families were first identified in the *Hepatocystis* reference assembly and are unique to *Hepatocystis* with unknown function [3]. Of the 16 genes in the *Hep1* gene family, we identified three as hypervariable in pHep and one as hypervariable in cHep. Of the 10 genes in the *Hep2* gene family, one was hypervariable in both pHep and cHep (HEP_00480100).

The same 1kb window spanning the circumsporozoite-related antigen gene (*CRA*; HEP_00212100 in *Hepatocystis*) was hypervariable in both pHep and cHep. *CRA* is conserved across *Plasmodium* and is involved in hepatocyte invasion [35,36]. In cHep, multiple bins spanning a gene encoding the oocyst capsule protein (*Cap380*, HEP_00340200), were found to be among the most hypervariable. In *Plasmodium, Cap380* is essential for survival and development of *Plasmodium* oocysts into sporozoites within the mosquito *[37]*. We also identified a hypervariable bin spanning a gene encoding an early transcribed membrane protein (*ETRAMP13*, HEP_00510800; Fig 5 and S5 Table). *ETRAMPs* are mostly expressed during asexual replication in the red blood cells in *P. falciparum*, but orthologues in *P. berghei* are expressed during liver stage development [38] Although conserved across *Plasmodium* species [36–38], *Cap380*, *CRA*, and *ETRAMP13* genes were not found to be hypervariable in our analysis of *P. falciparum*. In fact, in the 30X *P. falciparum* dataset, there were no hypervariable genes that were also hypervariable in *Hepatocystis*.

## No evidence for association between *ACKR1* genetic variation and *Hepatocystis* infection in *Chlorocebus*

In humans, a SNP in the GATA-1 transcription factor binding region upstream of the *ACKR1* gene (previously known as *DARC*) is associated with protection against *P. vivax* (rs2814778, -67 T>C) [39,40]. Similarly, it has been suggested that a SNP in the 5'UTR of the *ACKR1* gene in *Papio cynocephalus* may reduce susceptibility to *Hepatocystis* infection (A>G 359 bp upstream of the transcription start site) [41]. Using the inferred infection status and host genetic variation data, we sought to test for association between *ACKR1* variation and *Hepatocystis* infection in this dataset. Since the *Papio* samples had a lower average host coverage than the *Chlorocebus* samples (1.5X vs 5X), we tested for association in *Chlorocebus* using publicly available variant calls [42]. When considering all *Chlorocebus* samples, we identified five nonsynonymous SNPs within the *ACKR1* gene, including two within extracellular domains, and six within 1,000 bp upstream of the *ACKR1* 5'UTR that could potentially be involved in regulation of *ACKR1* and encompass the regulatory region where associations have been reported. However, when only considering SNPs segregating in countries where infections were present, none were significantly associated with *Hepatocystis* infection (S6 Table). We observed three missense variants, one synonymous, one intronic and one upstream variant that were present in uninfected samples and absent from infected samples, potentially consistent with a protective effect. However, larger samples will be required for sufficient power to evaluate this (S6 Table).

## Discussion

In this study, we analyzed publicly available whole genome sequence data to survey the distribution of *Hepatocystis* in 326 wild NHPs from Africa and Asia. Using Kraken2 for read classification allows many closely related genomes to be evaluated at once, with unique kmers supporting evidence for presence of sequences from a particular species. We then followed this with competitive mapping to a specific joint host and parasite reference genome, using both nuclear and

mitochondrial genome coverage to identify samples with elevated values for all three metrics. We found these were correlated and pointed to a set of individual NHPs that had been sampled while infected with *Hepatocystis* (Fig 2A).

Of the 17 species of NHPs included in this dataset, five do not have published surveys for *Plasmodium* or *Plasmodium*-like infections [43]. These species are *Pongo abelii*, *Theropithecus gelada*, *Macaca thibetana*, *Chlorocebus cynosuros*, and *C. tantalus*. However, we note that it is possible that previous surveys were negative and the findings were not published. Alternatively, they could have been surveyed under different species names. For instance, *Chlorocebus* were formerly considered to be in the genus *Cercopithecus* [44,45]. Similarly, *P. abelii* was not classified as a species until the mid-2000s [46]. Thus, if we assume they have not been previously sampled, *C. cynosuros*, could be a possible new host of *Hepatocystis* as we inferred an infection in this species. This also suggests *Hepatocystis* is circulating in the Kafue region of Zambia where this individual was sampled from. We additionally report *Hepatocystis* infections in The Gambia for the first time, in eight of 22 *C. sabaeus* samples, highlighting the utility of surveying parasites in wild NHP whole genome sequence data to continue improving our understanding of the distribution of *Hepatocystis* or other blood-borne pathogens.

We also identify infections in species that are known hosts of *Hepatocystis*: *Papio cynocephalus*, *Chlorocebus aethiops,* and *Chlorocebus pygerythrus* [6,20–22,47]. In this dataset, these samples were collected from locations in Ethiopia, Kenya, and South Africa. Ethiopia has the highest geographic representation, comprising 25% of all samples in the dataset including *Papio hamadryas* (n = 20), *C. aethiops* (n = 16), and *Theropithecus gelada* (n = 48) [42,48]. Of the Ethiopian samples, *C. aethiops* was the only species inferred to be infected with *Hepatocystis* in our analysis, consistent with previous surveys of the *Chlorocebus* genus and *P. hamadryas* in Ethiopia [20,22]. The higher altitude environment at which *T. gelada* lives may drive the absence of *Hepatocystis* [48], but other environmental factors, species-barriers, or vector preferences could also be responsible especially given the absence of *Hepatocystis* in *P. hamadryas* as well. Additional sampling would be needed to confirm absence of *Hepatocystis* in *Theropithecus* and *Papio* species in Ethiopia.

In this study, we identified *Hepatocystis* infections in *Papio cynocephalus* from Kenya and in five different taxonomic groups of *Chlorocebus*, each sampled from a different country: *C. aethiops* from Ethiopia, *C. cynosuros* from Zambia, C. *pygerythrus hilgerti* from Kenya, *C. pygerythrus pygerythrus* from South Africa and *C. sabaeus* from Gambia. Phylogenetic analysis of *cytb* sequences and principal components analysis based on genome-wide variation suggest that parasite population structure follows host species rather than geography. In Kenya, *P. cynocephalus* and *C. pygerythrus hilgerti* carried *Hepatocystis* infections, but these do not cluster together in either the *cytb* phylogeny (Fig 3) or PCA (Fig 4A). Instead of grouping by geography, we found that the *cytb* sequences from *Chlorocebus* all cluster together, and that PC1 of all samples together separates cHep from pHep. A lack of geographic clustering has been observed among the *Hepatocystis cytb* sequences sampled from six primate host species within a single national park in Uganda, which instead cluster by host species [24]. It is possible that these represent different species of *Hepatocystis*, but this determination requires additional considerations such as connecting sequence data to morphology, quantification of genetic divergence, and/or multilocus phylogenetic analysis.

The sampling of identical *cytb* sequences within a single *Chlorocebus* species but not between them suggests additional substructure. This is also strongly supported by PCA of cHep only, where the first PC reveals three groups of parasites infecting *C. aethiops* in Ethiopia*, C. sabaeus* in the Gambia*,* and the remaining three *Chlorocebus,* which share a complex demographic history. Published analysis of *Chlorocebus* host genomic data did not support a closer relationship between *C. pygerthrus hilgerti* and *C. pygerythrus pygerythrus* than either has with *C. cynosuros*, and their status as species or subspecies remains debated [42]. In our analysis of genomic structure by PCA, *Hepatocystis* sampled from these three taxonomic groups were genetically similar, despite geographic distance, compared with those sampled from *C. aethiops or C. sabaeus*, suggesting that parasite genetic structure may mirror the close relationships among these hosts.

In both cHep and pHep we find evidence for diversifying selection on genes in two *Hepatocystis*-unique gene families, *Hep1* and *Hep2*. Although their function remains unknown, the expansion of both families and high SNP density suggest they

may be involved in immune evasion. It is possible that higher copy number than represented in the reference genome could contribute to the signal of high SNP density, although we note that the signal is present even though high depth sites were removed from the analysis. Several *Hep1* and *Hep2* genes are highly expressed in blood stages, although unlike *Plasmodium*, *Hepatocystis* does not undergo asexual replication in the blood stage [3]. Despite lacking this stage and the associated pathogenic outcomes, the *Hep1* and *Hep2* gene families may function in immune evasion similar to other hypervariable gene families across *Plasmodium* (e.g., *var*, *SICAvar*, *pir*). *CRA*, a single copy gene with a role in hepatocyte invasion, also has high SNP density in both cHep and pHep. Finally, the gene *Cap380*, which is involved in oocyst development into sporozoites within the insect vector, shows the highest SNP density in the genome in cHep but is not hypervariable in pHep, possibly resulting from interaction with a different *Culicoides* vector species. However, we note that differences between high SNP density windows in cHep compared to pHep could also be due to limited sensitivity in low coverage data.

Paired with host variation data, infection inference allows for the potential discovery of SNPs associated with *Hepatocystis* infection. Although the current sample size and host genomic coverage limit this application, we demonstrate this potential by testing for association between *Hepatocystis* infection and variation in the *ACKR1* gene, which has previously been associated with *Hepatocystis* infection in baboons (genus *Papio*) [41]. We identified six alleles present only in uninfected *Chlorocebus* samples, but none were significantly associated with infection status. Additionally, two of these are missense variants in extracellular regions, although not within the region where *P. vivax* is thought to bind to the receptor in humans [49]. Future studies with larger sample sizes and including baboons could shed light on whether there is consistent evidence for association between *Hepatocystis* and *ACKR1* variation in NHPs, paralleling the association between *ACKR1* and *P. vivax* in humans. However, it is unknown whether *Hepatocystis* uses ACKR1 as a receptor or if *Hepatocystis* has been a strong selective pressure given the milder disease manifestation.

In this analysis, we did not find any confidently inferred *Plasmodium* infections upon combining Kraken2 read classification and coverage across the nuclear and mitochondrial genomes. This is likely a sampling bias rather than any inherent difference in detectability using our pipeline, as African great apes, hosts of multiple close relatives of human *Plasmodium* species, and macaques, hosts of at least four species of *Plasmodium*, were underrepresented in our study. Two species of *Plasmodium* have been found in African monkeys, *P. gonderi* and *P. mandrilli*, although largely in species for which WGS data was not available [50–52]. Increased sampling from wild NHP populations would therefore be an opportunity to identify *Plasmodium* infections as well.

We were able to infer *Hepatocystis* infections in samples with as low as 0.4X host genome coverage, suggesting this approach is applicable even to low-coverage NHP sequencing projects as they become available. Nonetheless, lower coverage of both host and parasite genomes limits the sensitivity of detection and we may miss true infections. An additional challenge and potential bias is a more limited ability to detect or map reads that are more divergent from the included reference genomes. Additionally, due to low coverage, our analysis was limited to SNP variation. Detection of more complex variant types and the application of tests for selection requiring confident individual-level genotype data were not feasible with this dataset. We attempted to run a similar pipeline on samples from fecal and non-blood tissue samples, which would broaden the opportunities for parasite identification to non-invasively collected samples. However, we found much noisier read classification in these sample types and additional methods development may be required for this application.

Overall, we demonstrate that shotgun sequences derived from whole blood can be used to identify apicomplexan parasites and their distribution across NHP populations. We describe evidence for *Hepatocystis* population structure between host primate species supported by variation across the nuclear genome. As no other genome-wide population samples are available, interpretation is limited to relationships among the samples within this study. Wider implementation of these approaches will enhance our ability to study parasite evolution and host-parasite relationships as sequencing data from more wild populations becomes available [53,54]. Notably, this approach can be adapted to study infections and associations in other vertebrate hosts of *Plasmodium*, invertebrate vectors, or even expanded to other host-parasite systems that may be difficult to sample in the wild.

## Materials and methods

### Data curation

Since apicomplexan parasites invade red blood cells as part of their lifecycle, DNA from *Hepatocystis* or *Plasmodium* in infected wild primates can be recovered in raw shotgun DNA sequences obtained from whole blood samples. We therefore searched the NCBI SRA database, which is a repository for raw shotgun DNA sequence data and metadata on sampling location and source material, (https://www.ncbi.nlm.nih.gov/sra) to compile a dataset of publicly available, wild, non-human primate sequences from blood samples. Similar to Hernandez et al. 2020 [55], we performed a search for "(primate OR primates) AND (genome or genomic) NOT (Homo sapiens)". We also did a specific search for each recognized non-human primate species, as listed in S1 Table from Hernandez et al. 2020, e.g., "Macaca fascicularis AND whole genome". Search results were then filtered to only include samples described as "wild" and where a geographic location was provided. Paired FASTQ files for the 326 samples meeting these criteria were downloaded using sra-toolkit [56] (S1 Table). The search covered sequences available as of August 31st, 2022. We note that the high coverage *Piliocolobus tephrosceles* sequence data from which the *Hepatocystis* reference genome was assembled is not publicly available and thus was not included in this dataset.

### Taxonomic classification with Kraken2

We created a custom Kraken2 database using the steps listed in the online manual section 9 (https://github.com/DerrickWood/kraken2/wiki/Manual) and included the following libraries: Protozoa, Archaea, Bacteria, and Virus [27,28]. The standard Protozoa library included with `kraken2-build` contains 13 *Plasmodium* reference genomes. These are: *P. vivax*, *P. malariae*, *P. cynomolgi strain B*, *P. knowlesi strain H*, *P. chabaudi chabaudi*, *P. yoelii*, *P. vinckei vinckei*, *P. berghei ANKA*, *P. sp. gorilla clade G2*, *P. reichenowi*, *P. gaboni*, *P. coatneyi*, and *P. relictum*. Because contamination from adapter, vector, or primer sequences can be found in sequence data and reference genomes, we also included the decontamination library, Univec [57]. Three primate reference genomes were also included to assess the number of reads classified as primate origin for each sample, allowing us to later compare the proportion of parasite relative to primate reads and to prevent any primate sequences from being incorrectly classified as parasite sequences. The primate genomes included were:

- Human GRCh38 (https://ftp.ncbi.nlm.nih.gov/genomes/all/GCF/000/001/405/GCF_000001405.40_GRCh38.p14/)

- Rhesus Macaque Mmul10 https://ftp.ncbi.nlm.nih.gov/genomes/all/GCF/003/339/765/GCF_003339765.1_Mmul_10/

- Gray Mouse Lemur Mmur_3.0 (https://ftp.ncbi.nlm.nih.gov/genomes/all/GCF/000/165/445/GCF_000165445.2_Mmur_3.0/)

We additionally added the *Hepatocystis* reference genome to the Kraken2 database (https://ftp.ncbi.nlm.nih.gov/genomes/all/GCA/902/459/845/GCA_902459845.2_HEP1/).

We then ran Kraken2 on all sequence runs from the 326 samples identified in the data search as well as paired FASTQ files from the 10 French individuals in the Human Genome Diversity Project shotgun sequencing data [58]. The French HGDP population served as a negative control since this population is not expected to be infected by *Plasmodium* or *Hepatocystis* parasites. For samples with multiple runs, each set of paired FASTQ files were input separately to Kraken2 and then combined to the sample level for downstream analysis. The `--report` flag was included to generate the standard Kraken2 output as well as a summarized classification file for downstream analysis.

### Inference of individual infection status

Three criteria were used for inferring infection status: the proportion of reads classified as parasite, the mtDNA coverage, and the nuclear coverage.

For each sample, we calculated the proportion of reads classified from each *Plasmodium* and *Hepatocystis* species individually, by dividing the number of reads classified as the specific parasite species by the sum of reads classified as any primate taxon and the number of reads classified as that parasite species, in other words:

$$P_{ij} = \frac{\text{\# reads from parasite species } i \text{ in sample } j}{\text{\# reads from any primate} + \text{parasite species in sample } j}$$

We considered samples where the proportion of reads classified as a parasite species was more than six standard deviations greater than observed in the HGDP French population as potential infections. To further confirm positive infections in these samples, we used BWA-MEM to align the sequences to a merged primate and parasite reference genome for the specific primate and parasite pair using the closest available reference genome [59]. For each sample, coverage of the parasite genome was calculated using mosdepth [60] in order to determine mean mitochondrial and nuclear coverage. We used cutoffs of>0.1x for nuclear DNA coverage and>1x for mtDNA coverage to consider a sample infected. This was based on both the observed increase in coverage in samples with higher proportion of parasite-mapped reads (Fig 2A) and typical values sufficient for ultra-low and low-coverage analyses [61–63].

## Phylogenetic analysis of *cytochrome b*

To assemble the *cytochrome b* (*cytb*) sequence from each infected sample, we extracted read pairs that mapped to a 1.1kb region of the *Hepatocystis cytb* gene (HEP_MIT003, LR699572.1:5426–6550). FASTQ files containing the extracted reads were input for assembly with SPAdes [64] with a requirement of at least two reads covering each position (coverage >= 2X). We used the reference sequence as input for the `--trusted-contigs` flag to improve assembly of the gene region. SPAdes assembly also fails when coverage in the region is not sufficiently uniform. Successful *cytb* assemblies were obtained for 20 of the 37 samples.

To compare the sequences with previously published phylogenies, we downloaded all available *Hepatocystis cytb* sequences, sourced from three published studies [17,18,23–25] and one unpublished study (NCBI Popset 293411005) from the NCBI nucleotide database. We also included a sequence isolated from an Asian short-nosed fruit bat as an outgroup [15]. For the two macaque-infecting datasets, we used just two macaque-infecting sequences from each [17,18] given the high sequence similarity among the sequences. Additionally, in the Ayouba et al. dataset [23], we limited sequences to the unique haplotypes as described in their results. All sequences were aligned with MUSCLE [65] and trimmed using GBLOCKS, within SeaView [66], to address the differences in regions sequenced across studies. This resulted in a 727 bp sequence alignment with 84 unique sequences, containing 76 downloaded and the eight from our assemblies. The resulting multiple sequence alignment was exported in PHYLIP format and used as input for the IQ-TREE web interface to infer a phylogenetic tree using maximum likelihood [67–69]. We used IQTREE ModelFinder to select the best nucleotide substitution model (TIM2 + F + G4). Bootstrap branch support was performed with UltraFast with 1000 replicates and the root was specified as an Asian fruit bat sequence (FJ168565.1).

## Analysis of genome-wide genetic variation in *Hepatocystis*

Given that per sample coverage of the *Hepatocystis* nuclear genome was low (average coverage 0.32X), we used ANGSD [30] to assess genetic variation in a probabilistic framework using genotype likelihoods. From the alignments to a merged primate and *Hepatocystis* reference genome, we extracted the properly paired reads that aligned to *Hepatocystis* into a separate BAM file for each sample. To account for possible inclusion of reads from *P. gonderi,* known to infect African old world monkeys [50] but absent from the Kraken2 database, these reads were remapped to a joint reference including both *Hepatocystis* and *P. gonderi* contigs. A minority of reads mapped to *P. gonderi* (mean 1.5%, range

0.6-3.3%) and were excluded from further analyses. Reads mapping to *Hepatocystis* were again extracted as per-sample BAM files and used as input for SNP calling using the ANGSD SAMtools model (`-GL1`). This outputs allele frequencies for the major and minor alleles (`-doMajorMinor 1`, `-doMAF 2`) at inferred variable sites as well as individual genotype likelihoods for these sites. Only biallelic variants with a significant P-value (p < 1e-06) were kept. We generated three variant call sets using different sets of sample BAM files as input: only *Hepatocystis* from *Papio* hosts, only *Hepatocystis* from *Chlorocebus* hosts, and *Hepatocystis* from all hosts.

To identify hypervariable regions of the genome, we calculated SNP density in 1 kb windows separately in the cHep and pHep variant calls. We excluded sites with very high coverage (top 0.1% per site coverage) to reduce the inclusion of regions with increased copy number relative to the reference sequence.

*Reference genomes used for joint mapping with Hepatocystis:*

- *Papio anubis*: https://ftp.ncbi.nlm.nih.gov/genomes/all/GCF/000/264/685/GCF_000264685.3_Panu_3.0/GCF_000264685.3_Panu_3.0_genomic.fna.gz

- *Chlorocebus sabeus*: https://ftp.ncbi.nlm.nih.gov/genomes/all/GCF/015/252/025/GCF_015252025.1_Vero_WHO_p1.0/GCF_015252025.1_Vero_WHO_p1.0_genomic.fna.gz

- *Plasmodiun gonderi*: https://ftp.ncbi.nlm.nih.gov/genomes/all/GCA/036/584/725/GCA_036584725.1_Pgonderi_v2_iGEM/GCA_036584725.1_Pgonderi_v2_iGEM_genomic.fna.gz

To assess how well we could capture genomic variation with the low coverage data, we applied the same ANGSD pipeline to 25 publicly available high coverage *P. falciparum* genomes [70] subsampled with SAMtools [71] to compare a standard high coverage of 30X with two low coverages: 1X and 0.5X. We considered the top 1100 windows (> 2SD from the mean SNP density) as true positives in the 30X data. We then plotted precision and recall using cut-offs for true positives in the low coverage datasets ranging from 10 to 100 in increments of 10 (S4 Fig).

For PCA analysis, we ran pcangsd [33] to estimate a covariance matrix using genotype likelihoods (i.e., beagle files). As input we used either variants called from cHep, pHep, or cHep and pHep combined. The PCs were then calculated in R using the eigen() function and the percent of variance explained was calculated by (eigenvalue/ total sum of eigenvalues) *100.

## Association of ACKR1 variation with infection status in Chlorocebus

To transfer annotations from the orthologous regions of the human gene model, we used BLAT [72] from the UCSC genome browser with the sequence of the canonical human *ACKR1* transcript (ENST00000368122.4/NM_002036.4) as the query and the *Chlorocebus sabeus* reference genome (chlSab2). We then annotated the location of the GATA1 binding region and the human SNP within it encoding the Fy$^{ES}$ (Duffy null) allele (rs2814778) and the SNP determining the Duffy A/B blood group polymorphism (rs12075) onto the corresponding location in the *C. sabeus* sequence.

We extracted variant calls for *Chlorocebus* samples included in our dataset in a 2.6 kb region (chr20:4742279 – 4744887) containing the *ACKR1* ortholog from the publicly available VCF file of biallelic SNPs [42]. Variant annotations were added with SnpEff [73]. There were 46 SNPs in this region among the 23 infected and 140 uninfected samples. Samples from countries without any infections were excluded from association testing, resulting in 23 infected and 84 uninfected individuals from four countries (S6 Table). The dataset of SNPs within samples from countries with infections was further filtered to only include polymorphic sites with a minor allele count > 5, leaving 18 SNPs. Logistic regression was used to test for association between infection status and genotypes at each of the 18 SNPs under an additive model in R v4.2.3 using the lme4 (1.1-30) package. Country was included as a random effect to capture both country and species since each country had a single species.

PLOS Pathogens

## Supporting information

**S1 Fig. Percent of reads classified as A) *Hepatocystis* and B) *P. cynomolgi* by Kraken2 for all 326 samples.** Samples inferred as infected are colored purple, while those considered uninfected are shown in gray. Samples are grouped by host genus.
(TIF)

**S2 Fig. Proportion of sites at each coverage depth for each sample considered as infected.** (A) mtDNA contig in cHep. (B) mtDNA in pHep. (C) nuclear contigs in cHep. (D) Nuclear contigs in pHep.
(TIF)

**S3 Fig. Phylogenetic tree of *cytochrome b (cytb)* sequences using only sites present across all sequences in the alignment.** The maximum likelihood tree was constructed with IQ-TREE from the 20 *cytb* sequence assemblies in our dataset together with the *Hepatocystis* reference sequence and additional 75 unique publicly available *Hepatocystis* sequences from various NHPs sequenced via PCR (alignment length = 287 bp, minimum sequence length = 287 bp, sites present in all sequences = 287 bp). Tips representing sequences from our dataset are colored and labeled with the host species name and number of samples carrying the same sequence. Tips from published studies are labeled by host genus and a sample ID. Bootstrap values >50% are indicated with an asterisk. Color key: red – outgroup from bat host. Purple – *Chlorocebus* hosts from our study. Dark green – *Papio* hosts from this study. Light green – *Papio* hosts from other studies. Orange – Asian primates (macaques). Blue – the reference sequence. Black – African primates.
(TIF)

**S4 Fig. Precision recall curve for SNP densities in the *P. falciparum* dataset using hypervariable bins in the 30X data as truth sets, defined as bins with SNP density greater than 2 standard deviations (1 SD = 0.035) from the mean (mean = 0.017).** We then used 10 thresholds for classification as hypervariable in the low coverage data. These ranged from 10 to 100 bins in increasing in increments of 10, shown as points on the plot. Points shaped as stars represent the threshold of 50 bins.
(TIF)

**S1 Table. Sample information for each BioProject in the curated dataset including species, sampling locations, tissue type, number of individuals, NCBI BioProject ID, and the study DOI or Grant ID.**
(XLSX)

**S2 Table. Per-sample information and metrics used for inferring *Hepatocystis* infections in *Papio* and *Chlorocebus* samples.** This table includes the SRA ID, host genus and species name, country of sampling, % of reads classified as *Hepatocystis* by Kraken2, the mean nuclear and mitochrondrial coverage for *Hepatocsytis* as well as the ratio of these coverages, and whether or not the sample was inferred as infected. For infected samples, the host genome coverage and whether cytb assembly was successful are also shown.
(XLSX)

**S3 Table. The total number of samples and the number of *Hepatocystis*-infected samples for each species by sampling location.**
(XLSX)

**S4 Table. Hypervariable genes identified in comparable analysis of *P. falciparum*.** Genes containing hypervariable bins identified in both the 1X and 30X *P. falciparum* datasets. The columns are: gene ID, gene annotation, and the number of hypervariable 1kb bins overlapping each gene.
(XLSX)

**S5 Table. Genomic coordinates and gene content of the 50 most hypervariable 1 kb bins identified in pHep.**
(XLSX)

**S6 Table. Genomic coordinates and gene content of the 50 most hypervariable 1 kb bins identified in cHep.**
(XLSX)

**S7 Table. Logistic regression results for the *Chlorocebus* variants segregating in samples from countries where infections were inferred.** The odds ratio and 95% confidence interval, p-value, allele annotations, frequencies, and counts are included.
(XLSX)

## Acknowledgments

We gratefully acknowledge the support and resources from the Center for High Performance Computing at the University of Utah, especially Brett A. Milash who helped to set up the initial Kraken2 database used in this analysis. We thank Nels Elde, Sarah Bush, Aaron Quinlan, and Timothy Webster from the University of Utah for their guidance and feedback on this work.

## Author contributions

**Conceptualization:** Paige E Haffener, Ellen M Leffler.

**Data curation:** Paige E Haffener.

**Formal analysis:** Paige E Haffener, Helena D Hopson, Alejandra Herbert-Mainero, Anayansi Ramirez.

**Funding acquisition:** Ellen M Leffler.

**Investigation:** Paige E Haffener, Helena D Hopson, Alejandra Herbert-Mainero, Anayansi Ramirez.

**Methodology:** Paige E Haffener, Helena D Hopson, Ellen M Leffler.

**Supervision:** Ellen M Leffler.

**Visualization:** Paige E Haffener, Helena D Hopson, Alejandra Herbert-Mainero, Anayansi Ramirez.

**Writing – original draft:** Paige E Haffener, Helena D Hopson, Ellen M Leffler.

**Writing – review & editing:** Paige E Haffener, Helena D Hopson, Alejandra Herbert-Mainero, Ellen M Leffler.

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
