## [Decision Letter · Decision Letter 0]

Dear Dr Leffler,

Thank you very much for submitting your manuscript "Phylogenetics and genomic variation of two genetically distinct Hepatocystis clades isolated from shotgun sequencing of wild primate hosts" for consideration at PLOS Pathogens. As with all papers reviewed by the journal, your manuscript was reviewed by members of the editorial board and by several independent reviewers. In light of the reviews (below this email), we would like to invite the resubmission of a significantly-revised version that takes into account the reviewers' comments.

The manuscript has been seen by four reviewers. Most found it interesting, but have made many suggestions that could improve the manuscript. One reviewer was quite negative. My own view is that the manuscript is potentially interesting, but I have two main comments. First, I think there is a need to modify the presentation and interpretation of the phylogenetic analysis, toning down the conclusions. (More details below.) Second, to overcome the comments of the negative reviewer, I think there should be much more emphasis on the results obtained from having mined the numerous nuclear genome sequences.

We cannot make any decision about publication until we have seen the revised manuscript and your response to the reviewers' comments. Your revised manuscript is also likely to be sent to reviewers for further evaluation.

Sincerely,

Paul M. Sharp

Guest Editor

PLOS Pathogens

Francis Jiggins

Section Editor

PLOS Pathogens

Michael Malim

Editor-in-Chief

PLOS Pathogens

orcid.org/0000-0002-7699-2064

Phylogenetic analyses:

Figure 3: The tree in part A, where only the new sequences plus one other (the “ref” sequence) are included, is redundant. The text (line 170) refers to the length of this central branch (0.032) as being meaningful, but this is misleading. The impression of a long central branch separating two clades disappears when other sequences are included (as shown in part B).

The tree in part B should be rooted in a more appropriate position. As one reviewer suggests, this could be achieved by including outgroup sequence(s). The closest outgroups are likely to be Hepatocystis from bats. The tree could also be presented more clearly. It looks like every sequence is version 1, and so the labels do not need to indicate this. You could perhaps preface each label with a 2-letter code for the genus.

The misleading tree in part A is used to suggest there are two species here (i.e., cHep is one, pHep plus “ref” is the second). Even after the inclusion of other sequences in part B, the text continues to refer (line 184) to the pHep sequences (plus others from other hosts) as “this species”, but it is not at all obvious how this species is demarcated. It would not seem to make much sense to suggest, from the tree in part B, that there are two species, where the second includes all sequences beyond cHep. But how many species are there here? (I think it impossible to decide.)

When the tree in part B is rooted in a more appropriate position, the clade of cHep sequences will seem less special; not very different from the clade of exclusively Piliocolobus-derived sequences near the foot of Figure 3B.

In addition, at line 292 it is suggested that pHep clustered with isolates from 6 different genera. However, in Figure 3B, the majority of the pHep sequences fall in a cluster with (only) other Papio-derived sequences, plus a single chimpanzee-derived sequence; as reviewer #3 suggests, the chimpanzee-derived sequence may not be a real infection. Only one pHep sequence falls outside this cluster; the larger cluster including that other pHep sequence has no sequences from Mandrillus, Miopithecus or Colobus. In fact, I can only see single sequences from each of these three genera in the tree, and they are all clearly more closely related to cHep than to pHep.

Finally, it seems ironic that the Introduction refers to the limitations of cytb phylogenetic analysis (line 89), but then the only phylogenetic analysis in the paper is cytb.

In conclusion, I suggest deletion of Figure 3A, rooting the tree in Figure 3B more appropriately, and a revision of the conclusions derived from this phylogeny.

Other comments:

Figure 1: not quite “by taxonomic group”. The boxes above place the OWMs in two groups based on geography. Taxonomy would cluster the first five species in Cercopithecini, and the next six species in Papionini. Genus names should be given (“P.” stands for three different names here.)

Line 240: It would help to explain that ACKR1 was previously known as DARC.

Lines 333-335: African apes have also been found to be infected with relatives of P. vivax and P. malariae (and probably the two P. ovale species). African monkeys (of several species) have been found infected with P. gonderi and P. mandrilli (previously referred to as DAJ-2004).

Reviewer's Responses to Questions

**Part I - Summary**

Reviewer #1: In this important work, Haffener et al. expand the number of known genomic samples for the enigmatic parasitic genus Hepatocystis by two orders of magnitude, identifying reads captured as part of attempts to sequence primate genomes. In doing so they identify a new putative species of Hepatocystis, confined to certain host species, alongside other important insights. This is impactful work, which has been rigorously carried out and is well-described.

Back in 2018, this reviewer was involved in the first identification of a genome for Hepatocystis. This came about because in the process of sequencing a red colobous monkey, experimenters acquired sufficient reads that their final monkey assembly, deposited on NCBI, contained many contigs that were in fact from Hepatocystis. Puzzled by BLAST searches that reported that “monkey” genes were closely related to genes of the malaria parasite (https://theo.io/blog/2018/04/23/how-i-stumbled-upon-a-novel-genome-for-a-malaria-like-parasite-of-apes/), and we went on to characterise this genome in detail (https://www.ncbi.nlm.nih.gov/pmc/articles/PMC7425995/). That finding was the result of serendipity, and made possible by the large amount of Hepatocystis present in the sample.

Here, the authors instead take a targeted approach, they download the raw reads captured as part of a large number of primate genome sequencing efforts and look for samples containing Hepatocystis, by matching kmers to that initial genome (and also using other approaches). They survey 326 primate genomes and identify 30 Hepatocystis infections that have left significant genomic material, increasing the number of known genome-scale datasets of Hepatocystis by more than an order of magnitude. Although many (or all) samples have very low coverage, they are able to make use of these datasets in two main ways. Firstly, because mitochondrial DNA is much more abundant than nuclear DNA, they are able to accurately infer sequences for the samples' cytochrome B genes and infer a phylogeny. This phylogeny clusters by host species, with Chlorocebus-infecting Hepatocystis genomically distinct from those infecting baboons or colobus monkeys, the former representing a putative new species. Secondly, even though the samples' coverage is too low to be confident of the base at any particular position in a particular sample, the authors use a probabilistic approach to identify regions of the genome that show the most variation between samples. Their results here are plausible in terms of what we know about gene function in Plasmodium. Additionally, wherever the authors use new methodology they perform rigorous benchmarking using simulations based on malaria samples to demonstrate whether it has good power. 

All of the claims made in the manuscript are well-demonstrated and carefully described. The identification that Hepatocystis sequences are widespread, of a new species with limited host range, of the lack of apparent species-specificity by geography, and of apparently highly variable genes in Hepatocystis are all important discoveries, as is the proof-of-concept that this work offers for its methodology.

- Theo Sanderson

Reviewer #2: The manuscript of Haffener et al. uses a bioinformatics approach to scour published genome sequencing projects of OWMs for sequences deriving from Hepatocystis parasites, followed by phylogenetics to investigate evolutionary relationships among putative Hep lineages and analyses of SNP variation to investigate genes putatively involved in adaptation. Overall, the results are potentially very interesting as little is known about the diversity and distribution of Hepatocystis parasites infecting OWMs. The text is well-written, and the methods are mostly clearly explained. The figures are nicely presented but some require additional clarification regarding what exactly is being shown. I think the core methodological approach employed is robust, although I was uncertain about some of the downstream methods used to arrive at their final number of ‘true’ infections, which I found to be perhaps overly complicated. A couple of key issues that I would like to see addressed before consideration for publication are outlined below.

Reviewer #3: This study provides crucial insights into the evolutionary history of parasites from the Hepatocystis genus, which are widely distributed among mammals, including rodents, bats, and primates. Nevertheless, the phylogenetic relationship between this genus and parasites of the Plasmodium genus remains a topic of ongoing debate. The findings presented in this study significantly enhance our understanding of these parasites. However, the impact of this work could be further amplified by refining a few minor aspects, which, in my view, would elevate the overall quality of this already excellent research.

Reviewer #4: The genus Hepatocystis remains understudied, largely due to difficulties to sample and monitor infections in wild hosts, incl. non-human primates (NHPs). The present study presents a dark-mining approach of publicly available shotgun sequence reads of NHP blood samples. Accordingly, the authors have not generated any original data, but merely assembled available data. They searched 326 samples from 17 NHP species and considered appr. 10% (30) samples infected with Hepatocystis, based on a cut off of 0.002% 'Hepatocytis-like' sequence reads. The vast majority of the samples (28) had reads < 0.1%; only two samples were a little higher. This can be attributed to the overall low parasite abundance due to lack of erythrocytic schizogony. What is the advancement over a similar study (Ref.3)? Perhaps another proof that dark-mining works, perhaps a slightly improved molecular phylogeny, and/or perhaps a first glimpse at the (expected) genomic variation. The latter is not surprising, and a finding of interest to a very limited research community. Fig.3 displays the molecular phylogeny of the few NHP Hepatocystis genome sequences, which, of course, vary a bit. No context to other Hepatocystis sequences can be provided, which is, of course, due to lack of reference data. The SNP density (Fig. 4) gives a first, albeit preliminary, impression of SNP frequencies. Sequence assemblies and SNP analysis are sound, but the study lacks any salient finding that advances our understanding of Hepatocystis-host interactions, parasite stage conversion or infection dynamics.

**Part II – Major Issues: Key Experiments Required for Acceptance**

Reviewer #1: I believe the study's conclusions are already well justified.

Reviewer #2: General comments:

(1) The main pipeline uses Kraken2 to identify putative sequencing reads as deriving from Hepatocystis (or the primate host, or other Plasmodium) based on comparison of kmers (i.e., sequence identity). This seems like a sensible approach that I have no qualms about. However, I do wonder if the authors have sufficiently dealt with the potential problem of misidentification of sequences that in fact are from Plasmodium infections of OWMs, rather than Hepatocystis. Did the authors check the GC content of putative Hep reads? This should be distributed similarly to that seen for the reference Hepatocystis genome (i.e., around 20% GC, somewhat lower than most other Plasmodium species) and would be a relatively straightforward sanity check on the data. I understand that the authors ran their pipeline but searching for Plasmodium reads, rather than Hepatocystis, and came up with a null result (e.g., line 146), but it wasn’t mentioned if P. gonderi (as the only published genome of OWM-infecting Plasmodium species) was included in the reference set here. All this may not make a difference but it might be worth double-checking – my main concern would be if mis-identified Plasmodium reads have slipped through the net and are contributing to some of the ‘hypervariable’ regions identified downstream.

(2) Discussions about topological groupings (e.g. evidence for cHep and pHep putative species) based on the phylogenies shown in Fig. 3 are, at current, slightly meaningless because the trees are not rooted; monophyly of one set of taxa with respect to another is not possible without rooting the tree. This would be relatively easy to solve by including orthologous sequences from a range of other Plasmodium species and redrawing the tree, rooting as appropriate. This would have the added benefit of seeing relationships among putative Hep samples within the broader context of Plasmodium evolution. I strongly recommend additional sequences are added to at least Fig. 3B.

Reviewer #3: (No Response)

Reviewer #4: No experiments were performed in this study. The authors searched available sequence information and assembled the data. Accordingly, no validation, such as parasite microscopy, immunofluerescence stainings or qPCR analysis can be done.

**Part III – Minor Issues: Editorial and Data Presentation Modifications**

Reviewer #1: I stress again that the claims of the paper are already well-demonstrated. I do not see any of the below issues as essential to address for publication.

As a reader I initially expected the authors' approach to, as one small part "rediscover" the infection of the red colobus monkey that we had previously identified. In fact their survey does not include any colobus genomes. It could be useful to describe why - whether they were intentionally excluded as the only relevant ones had already been described (this is entirely fair, but is just not mentioned), or whether this is the result of some filtering.

I support experimentation with new graphical approaches but I personally didn't find Fig 1. especially intuitive - I think I'd find a stacked bar chart with countries labelled more intuitive (but this is a matter of opinion).

Fig. 2: I'd suggest expanding the "IQR" label to indicate it refers to coverage

Fig. 3: It could be useful to include an outgroup, such as a malaria cytb in some figure to put the relationships into context?

The authors simply describe the mean coverage of their samples as being very low, leaving open the possibility that they might have number of high coverage samples (though glancing at the BAM files this does not seem to be the case) - it would be great to give a bit more detail on the distribution of coverage, providing some sense of for example whether there are further sequences of sufficient coverage to allow assembly. Given that the colobus discovery was so accidental, I had expected that a targeted survey might discover more high coverage genomes, so if that was not the case that result seems worth reporting. I'm also interested in the extent to which cHep samples could be pooled to give a reasonable assembly. A supplementary table including coverage for each sample could be nice.

Abstract: "low coverage whole genome sequence (lcWGS) data from 326 wild non-human primates (NHPs) in 17 genera." This suggests that all samples the authors analysed had low (host) coverage. Is that the case? If so is that because all wild-primate genomes are thus, rather than due to filtering specifically for low coverage sequences?

With your finding of hypervariability in the HEP families: I can imagine a scenario in which highly duplicated gene families could complicate mapping and generate spurious variability (but of course a finding of genuine variability is extremely plausible in such cases do, as with the var genes). I guess that's just worth bearing in mind. I'm not requesting a change to the text.

Sometimes you use "vHep" and I'm unclear what is meant there (sometimes it is contrasted with pHep and sometimes cHep)

You mention Papio and you mention baboons. It might be nice to draw out the synonymous nature of these to the reader sometime to help the ignorant (like me)

In my view, all researchers should try to share their code as much as possible to help others to understand the details of their methods. To the extent that the authors have code for their analyses it could be useful if they could share this on Github. The code need not be polished.

Reviewer #2: Specific comments:

Line 133. “For further evaluation, reads from all Papio and Chlorocebus samples were mapped to a joint primate and Hepatocystis reference genome” I think other Plasmodium genomes should be included in this mapping step, as well as the Hepatocystis reference and other primate genomes. The inclusion of Plasmodium sequences would act as ‘decoys’ (similar to mapping strategies in RNA-seq analyses) to filter potential true Plasmodium reads from the final set.

Line 141. It would be helpful if more information was provided in terms of the results from each ‘infected’ dataset, in terms of recovery of putative Hepatocystis sequences. Proportion and length of Hep reference genome with reads mapping to it? Number of Hep genes covered, fully/partially? Any BUSCO genes covered? Average coverage is fine but it would be good to have additional details of the actual results here to aid understanding of these initial data.

Line 161. Did you attempt to assemble the putative Hep reads into contigs, other than for cytB?

Line 145. “Although we applied the same pipeline to look for Plasmodium, we found no clear infections supported by multiple predictors.” Could you expand on what exactly was done here? Results of these analyses could potentially answer my point above.

Line 136. “K-means clustering using the ratio of mtDNA to nuclear coverage, a measure of coverage uniformity (interquartile range), and the percent of reads classified revealed two out of three clusters that corresponded to elevated values across the three predictors.” I don’t fully understand this approach. What is the justification for choosing these 3 ‘predictors’ (ratio mt/nuclear, IQR coverage and % reads classified by Kraken) over others? It seems unnecessarily complicated – why not simply choose reads which preferentially map to the Hepatocystis reference genome, over primate and other Plasmodium decoys?

Fig. 2A. (i) There is quite a lot of blank space in this plot, is it necessary to show all the negative results? It’s also not clear from the plot itself that the X-axis of the bar plots represents the 326 samples, which, unlabelled and grouped by host genus, make it appear as if there is equal sampling across all 6 genera. (ii) Why choose 0.002% as the cut-off? Is this the 6 SDs difference compared to the French HGDP ‘negative control’ mentioned in the Methods? Please clarify.

Fig. 2B. (i) What are the points? Are they the 42 samples with > 0.002% Hep reads identified in part A? Please clarify. (ii) It would be helpful to briefly remind readers what IQR is and how it is calculated when defining the Y-axis. (iii) Also, the colour of cluster B vs cluster C is very hard for me to differentiate on the plot.

Figure 3. What is the X-axis? Genomic position of the Hep reference genome? Please clarify on the figure.

Line 192. It is unclear exactly how the mapping was performed during SNP calling, even in the Methods (see later comment). Either way it might be helpful to have a quick recap of how these variants were generated written here.

Line 357. Did the SRA search not flag the sequencing project known to contain the Hepatocystis reference genome previously analysed by Aunin et al. (2020)? It doesn’t seem to be mentioned in any of the results presented, was it manually removed at a later stage? It might serve as a useful positive control for some of the later analyses.

Line 369. Which Plasmodium species were included in the Kraken2 database? Was the OWM-infecting species P. gonderi included? If not, it might be useful to include this species to double-check if any reads are flagged as Plasmodium.

Line 413. I don’t fully understand the approach for identifying samples with putative Hep infection, which seems overly complicated. The first 2 indicators are reasonable enough, but what is the justification for the IQR metric? Unless I am mistaken a higher IQR value corresponds to more variable coverage across the Hep reference genome – why would this correspond with ‘true’ infection? In addition, for the k-means clustering part, why k = 3? Is there a reason for choosing this value? I do wonder if this part could be simplified – why not just include all samples with > 0.002% Hep reads and a reasonably complete cytB gene? The clustering analyses seems to have removed 12 samples (n reduces from 42 to 30, line 138); you could reasonably include all 42 samples with the caveat that some may have limited evidence, but presence of a cytB (or other marker) clustering with other Hep sequences is more convincing to me.

Line 438. How exactly was mapping performed prior to SNP calling? Were reads from different samples assigned unique read group IDs and all mapped together, or within pHep and cHep subgroups, or where individual samples mapped separately? Please provide a bit more detail.

S2 Fig. What is shown on the X-axes? The 329 FastQ samples? Please clarify. Also, why group samples by cluster? This seems kind of circular, since some of these metrics are used to define the clusters, but also biologically meaningless, since cluster ID is based on technical properties of the sampling strategy (i.e. probably influenced by things like type of extraction kit used, blood sample protocol, sequencing depth, primers used, amplification protocol (or not), etc…) rather than anything to do with the host or parasite itself.

Minor comments:

- Line 149. “Samples are grouped by >host< genus.”

- Line 154. “pHep” and “cHep” need to be defined.

- Line 291. “isolates” should be unitalicized.

- Line 389. What is the “French HGDP population”? Please define/clarify.

- Line 408. How was the alignment performed? What tools/parameters?

- Line 418. Presumably this was done for each sample separately? Please clarify.

- Lines 424 and 436. Please specify which model was used to construct the distance matrix for tree building. Was model testing implemented? A more recently published phylogenetics tool like IQ-TREE is able to run model testing and tree building very easily.

Reviewer #3: Minors corrections

Introduction

The text is well-structured, moving logically from background information to the study's objectives.

The use of phylogenetic context sets the stage for understanding the significance of Hepatocystis in relation to Plasmodium.

The authors are thorough in referencing previous studies, providing a solid foundation for the research.

It effectively highlights the need for genomic data to better understand Hepatocystis, indicating a gap in current knowledge that this study aims to fill

Howover, to improve this introduction, it could benefit from a clearer research question, a more focused explanation of the study's significance, and a brief overview of the methodology. Additionally, balancing the background information with a stronger emphasis on the novelty and broader implications of the findings would make the introduction more compelling.

For examples, the authors could rewritte some sentences which very long

authors could simplify sentences to make them less complex

1. Line 58-60 remove the apicomplexan and use only Hepatocystis parasites or genus

2. Line 60 delete the most and consider only Notably and rewrite this senence. I suggest you for example: “Notably, Hepatocystis parasites do not undergo asexual replication in red blood cells, which is responsible for the symptoms of malaria, and are transmitted by midges of the Culicoides genus rather than by Anopheles mosquitoes

3. Line62-65 this sentces is very long, could rewrit that, please

4. Would the lack of data provided by studies of Hepatocystis in rodents and bats not constitute a lack of appreciation of the positioning or analysis?

5. Please rewritte this sentence is not clear, line 85-88

6. Is there really any proof that chimpanzees are infected with Hepatocysitis? Given that chimpanzees feed on small monkeys and that this has been observed in the faeces, wouldn't this be the consequence of this? Also, is there a study using blood that shows this?

7. Line 96 “spp” is not italicised, change it throughout the manuscript,

8. Line 107, “spp” spp is not italicised, change it throughout the manuscript

Results

Curation

1. I think that the clarification of the purpose of databases mentioned here, enhancing the relevance of the data to the study

2. There is some redundancy in the percentages provided for the sample distribution. For example, "Of the remaining samples, 12% were collected in the Caribbean Islands and the other 12% were collected in Asia" can be streamlined

3. While the text mentions the SRA database and BioProject, it could be more explicit about what kind of data or information these databases provide to reinforce their relevance to the study.

4. Line 125: The definition of the term 'OWM' is superfluous, given that it has already been provided in the introduction. There is no reason for it to be repeated here.

Hepatocystis infections in African Old World Monkeys

1. This section of the text is characterised by a high level of technical vocabulary and complex sentence structures, which may prove challenging for readers with limited familiarity with the subject matter. The text could be made more accessible to a wider audience by simplifying some sentences and breaking down complex ideas into more digestible parts.

2. For example, it would be beneficial to rewrite lines 129-131, as they currently contain multiple ideas that could be more effectively conveyed through the use of simpler sentences.

3. Lines 136–138 comprise a dense and complex sentence, combining multiple metrics and results into a single unit of information. The text could be made more accessible by breaking it down into smaller sentences or simplifying it.

4. For instance, lines 143-146 could be rewritten as follows: "Additional cases with elevated values for one or two predictors may also represent true infections, especially those with lower overall coverage."

5. Please clarify why the authors do not use spp when discussing Hepatocystis or Hepatocystis parasites. Do the authors consider that this is a single species of Hepatocystis, or are they discussing the genus hepatocystis?

Genomic variation in Hepatocystis

The present section comprises a number of lengthy sentences that may prove confusing to the reader. It is therefore recommended that the authors shorten some of them.

For example "We used precision recall curves to determine a threshold for considering windows of high SNP density (S3 Fig). Using 1kb bins in the 30X data with a SNP density greater than 2 standard deviations from the mean as the truth set, both 1X and 0.5X datasets had high precision (100% for the top 10 bins and >96% for the top 50 bins) and overlapped genes known to be hypervariable in P. falciparum (S3 Table).

Discussion

The discussion is well written, although it does exhibit some shortcomings. However, these do not detract from the excellent quality of the work as a whole. Nevertheless, addressing these issues would enable the discussion to be presented in a more comprehensive, accurate and relevant manner to a broader scientific audience.

1. The discussion makes general observations without providing a detailed analysis of the genetic differences between the newly identified Hepatocystis species and other known species. This lack of detail makes it challenging to ascertain the significance of the findings.

2. The functional implications of the genetic variations observed in Hepatocystis in comparison to Plasmodium have not been sufficiently explored. The discussion does not elucidate how these differences might affect the parasite's biology or its interaction with hosts. It would be beneficial for the authors to develop this aspect further in their discussion.

3. The discussion provides a cursory overview of the co-evolutionary dynamics between Hepatocystis and its NHP hosts, yet lacks a comprehensive analysis. This leaves a gap in our understanding of the evolutionary dynamics at play.

4. The following assumptions are made about host specificity: The supposition that a singular generalist Hepatocystis species may infect a range of diverse African NHPs is posited without substantial corroboration. This weakens the argument and fails to consider the potential for more complex host-parasite relationships.

5. It is notable that the analysis relies heavily on a single genome sequence, which may limit the generalisability of the findings. Furthermore, the discussion does not adequately consider how additional genomic data might alter current interpretations. It would be beneficial for the authors to take this into account for a more comprehensive assessment.

6. Additionally, the limitations of the methods used, such as potential biases in the taxonomic classification tool (Kraken2) or challenges in mapping reads, are not sufficiently discussed. This could lead to an overestimation of the results' reliability.

Reviewer #4: Due to the lack of original data the authors maximized their display items. Fig. 1 (the NHP samples) will typically be supplemental data, even in very specialized journals. Similarly, Fig. 2, which contains no central information and simply describes the set of 30 samples with low Hepatocystis sequence reads.

PLOS authors have the option to publish the peer review history of their article (what does this mean? ). If published, this will include your full peer review and any attached files.

**Do you want your identity to be public for this peer review?** For information about this choice, including consent withdrawal, please see our Privacy Policy .

Reviewer #1: **Yes: ** Theo Sanderson

Reviewer #2: No

Reviewer #3: **Yes: ** Larson BOUNDENGA

Reviewer #4: No
---

## [Editor Report · Decision Letter 1]

May 19 2025

PPATHOGENS-D-24-01357R1

Phylogenetics and genomic variation of Hepatocystis isolated from shotgun sequencing of wild primate hosts

PLOS Pathogens

Dear Dr. Leffler,

Thank you for submitting your manuscript to PLOS Pathogens. After careful consideration, we feel that it has merit but does not fully meet PLOS Pathogens's publication criteria as it currently stands. Therefore, we invite you to submit a revised version of the manuscript that addresses the points raised during the review process.

Please submit your revised manuscript within 30 days May 19 2025 11:59PM. If you will need more time than this to complete your revisions, please reply to this message or contact the journal office at plospathogens@plos.org. Please include the following items when submitting your revised manuscript:

We look forward to receiving your revised manuscript.

Kind regards,

Paul M. Sharp

Guest Editor

PLOS Pathogens

Francis Jiggins

Section Editor

PLOS Pathogens

Sumita Bhaduri-McIntosh

Editor-in-Chief

PLOS Pathogens

orcid.org/0000-0003-2946-9497

Michael Malim

Editor-in-Chief

PLOS Pathogens

orcid.org/0000-0002-7699-2064

**Additional Editor Comments:**

Dear Dr. Leffler:

Thank you for your revised manuscript. I am happy with the manner in which you have dealt with the various points raised by the reviewers.

However, in reading through this revised version, I was struck by a few points, particularly in regard to figures 3 and 4.

Figure 3:

1. As suggested, you have added an outgroup to the phylogenetic analysis (Figure 3). Looking at the trees in Ref.9 (Schaer et al., 2017), it seems that the sequence from Pteropus (FJ168565) would be less distant from the primate-derived sequences, than the sequence from Cynopterus (MW366842) that you have used. Perhaps it would be better to use this other, closer outgroup?

2. The second sequence in the tree is labelled Pan. OL691976. But the GenBank entry OL691976 is described as being from a human sample.

3. Sequences 7 and 8 in the tree lack accession numbers.

4. The legend to Figure 3 mentions an alignment length of 727 bp. However, the analysis includes many sequences from the Ayouba et al. (2014) paper, some of which are much shorter that this. For example, JQ070897 is only 518 bp; i.e., nearly 30% of the alignment length is missing. It would convey more accurate information if you indicated the number of sites present in all sequences in the alignment.

Figure 4:

5. You have removed Fig.2A from the first version of the manuscript. You have inserted Figure 4, which potentially suffers from the same problem that Fig.2A had, that is the lack of samples placed at intermediate positions in the Hepatocystis tree. Thus, again your conclusions about the evidence for structure by host species (e.g., at line 35-36 in the Abstract) could be biased by the particular samples included in the analysis.

In the case of Figure 4, you could not include many samples, because most do not have nuclear genome coverage. However, it seems that you could (should) include the reference sample from the Piliocolobus genome in this analysis.

Other minor comments:

6. L.313-314. Would it not be true to say that “all Chlorocebus species” were considered as Cercopithecus?

7. L.337. “P.” should be “C.” Also, I note that Svardal et al. (2017; your Ref. 42) did not show hilgerti and pygerythrus as closer to each other than to cynosuros; I wonder why you consider these two (only) as subspecies?

I hope you can address these points.

Yours sincerely,

Paul Sharp

**Journal Requirements:**

1) Please include the affiliation of the author (Ellen M Leffler) in the online submission form.

2) Please include the source details of the map found in (Figure 2B) in the legend of the figure.

**Figure resubmission:**
---

## [Editor Report · Decision Letter 2]

Dear Leffler,

We are pleased to inform you that your manuscript 'Phylogenetics and genomic variation of Hepatocystis isolated from shotgun sequencing of wild primate hosts' has been provisionally accepted for publication in PLOS Pathogens.

Best regards,

Paul M. Sharp

Guest Editor

PLOS Pathogens

Dominique Soldati-Favre

Section Editor

PLOS Pathogens

Sumita Bhaduri-McIntosh

Editor-in-Chief

PLOS Pathogens

orcid.org/0000-0003-2946-9497

Michael Malim

Editor-in-Chief

PLOS Pathogens

orcid.org/0000-0002-7699-2064
---

## [Editor Report · Acceptance letter]

Dear Leffler,

We are delighted to inform you that your manuscript, "Phylogenetics and genomic variation of Hepatocystis isolated from shotgun sequencing of wild primate hosts," has been formally accepted for publication in PLOS Pathogens.

Best regards,

Sumita Bhaduri-McIntosh

Editor-in-Chief

PLOS Pathogens

orcid.org/0000-0003-2946-9497

Michael Malim

Editor-in-Chief

PLOS Pathogens

orcid.org/0000-0002-7699-2064